# Metagenomic sequencing of post-mortem tissue samples for the identification of pathogens associated with neonatal deaths

Vicky L. Baillie [1,2] ✉, Shabir A. Madhi [1,2], Vida Ahyong[3] &
Courtney P. Olwagen[1,2]

Postmortem minimally invasive tissue sampling together with the detailed review of clinical records has been shown to be highly successful in determining the cause of neonatal deaths. However, conventional tests including traditional culture methods and nucleic acid amplification tests have periodically proven to be insufficient to detect the causative agent in the infectious deaths. In this study, metagenomic next generation sequencing was used to explore for putative pathogens associated with neonatal deaths in post-mortem blood and lung tissue samples, in Soweto, South Africa. Here we show that the metagenomic sequencing results corroborate the findings using conventional methods of culture and nucleic acid amplifications tests on post-mortem samples in detecting the pathogens attributed in the causal pathway of death in 90% (18/20) of the decedents. Furthermore, metagenomic sequencing detected a putative pathogen, including *Acinetobacter baumannii, Klebsiella pneumoniae, Escherichia coli*, and *Serratia marcescens*, in a further nine of 11 (81%) cases where no causative pathogen was identified. The antimicrobial susceptibility profile was also determined by the metagenomic sequencing for all pathogens with numerous multi drug resistant organism identified. In conclusion, metagenomic sequencing is able to successfully identify pathogens contributing to infection associated deaths on postmortem blood and tissue samples.

In 2019, there were approximately 5.2 million deaths in children less than 5 years of age, 46% of which occurred within the first month (neonates) of life[1]. Seventy-seven percent of deaths in children under-5 years of age occur in Africa and South Asia[2]. Traditionally, the cause of death (CoD) in children is modeled mainly using verbal autopsy and vital registration data[3], and only attributes the death to a single underlying condition. The use of postmortem minimally invasive tissue sampling (MITS) has been shown to provide granular understanding of the CoD in children, including providing biological evidence of pathogen specific causes of infection related deaths. Furthermore, use of

MITS together with other sources of information enables better characterization of the full causal pathway to death, including expanding to identify antecedent and immediate events leading to death, over and above the underlying CoD. We reported in 2019, using MITS, the dominant role (74%) of multidrug-resistant (MDR) bacteria, particularly *Acinetobacter baumannii* and *Klebsiella pneumoniae*, in the causal pathway of in-hospital neonatal deaths in South Africa[4]. These pathogens were only identified by conventional culture, since the commercial molecular assays used at the time did not target some of the organisms, including *Acinetobacter baumannii* and *Klebsiella pneumoniae*[4].

[1]South Africa Medical Research Council Vaccines and Infectious Diseases Analytics Research Unit, University of the Witwatersrand, Faculty of Health Science, Johannesburg, South Africa. [2]Wits Infectious Diseases and Oncology Research Institute, University of the Witwatersrand, Faculty of Health Science, Johannesburg, South Africa. [3]Chan Zuckerberg Biohub, 499 Illinois St, San Francisco, CA 94158, USA. ✉e-mail: Vicky.Baillie@wits-VIDA.org

Metagenomic next-generation sequencing (mNGS) has the potential to provide a more granular understanding of the role of different pathogens in the causal pathway to death, as it is not biased to only identifying pre-selected targets, and is able to identify viruses, bacteria, fungi, and parasites in a single assay[5–7]. In a study to determine the etiology of meningitis, encephalitis, or myelitis for which no pathogen was identified by routine microbiological testing, mNGS yielded identification of a putative pathogen in 22% (13/58) of cases[5]. Furthermore, mNGS allows for culture-independent analysis of the antimicrobial resistome, enabling characterization of antimicrobial resistant genes present in the sample[8]. The sensitivity of standard microbial culture is variable on ante-mortem blood and tissue sampling, and the antimicrobial susceptibility profile is usually delayed for 2 to 3 days after initiation of empiric antibiotic therapy for suspected sepsis[9,10]. Rapid characterization of the antimicrobial resistome on ante-mortem samples could assist with antimicrobial stewardship and potentially improve patient outcomes.

In this exploratory study, we evaluated the utility of mNGS in identifying pathogens associated with neonatal deaths. We tested post-mortem blood and lung samples collected by MITS, which had previously been tested by culture and select nucleic acid amplification tests (NAAT)[4].

## Results

### Study population

Between July 16th, 2015 and July 30th, 2016, MITS was undertaken in 153 (64.8%) of the 236 neonatal deaths who were eligible for enrollment. The leading underlying CoD included complications of prematurity (53%), complications of intrapartum events (15%) and congenital malformations (13%). Among the decedents that died of non-infectious causes (65/153; 42%), 16 were included in the current study as controls. The leading immediate CoD among the controls was "respiratory distress syndrome of newborn" (50%, n = 8/16). Furthermore, there was no notable organisms detected by culture of NAAT either on ante-mortem blood sample or post-mortem sampling; Supplementary Table 1. Eight-eight (57.5% of 153) deaths were attributed to being associated with an infectious cause either as the immediate or underlying cause; including 55% (n = 48/88) due to sepsis, 38% (n = 33/88) due to lower respiratory tract infection (LRTI), and 8% (n = 7/88) due to meningitis or encephalitis. Of the decedents with an infectious cause, 35.2% (n = 31/88) had residual paired lung tissue and blood samples available for mNGS analysis and were included in this study. Of these 31 decedents, a causative pathogen was identified through conventional ante- or post-mortem culture or on NAAT (post-mortem sample only) in 65% (n = 20) of the decedents. Whereas in the remaining 11 cases, there was insufficient evidence for the DeCoDe panel to identify a causative pathogen; Table 1 and Supplementary Table 2. The medical notes and antemortem blood culture results were unavailable for a single case where infection was attributed in the casual pathway to death. In the 20 decedents where a pathogen was identified as the infectious CoD, 70% (n = 14/20) had the organism identified on ante-mortem samples within 72 h of death, of which 93% (n = 13/14) were also identified on post-mortem cultures. In the remaining 6 decedents attributed to infections as the cause of death, the causative pathogen was identified on postmortem blood (n = 5) and lung (n = 6); Supplementary Table 2. In the decedents where no causative pathogen was identified, all the antemortem cultures and 64% (n = 7/11) of the postmortem cultures were negative; Supplementary Table 2.

The median time between the death of the neonate and the MITS procedures for the controls was 19.4 h (SD:10.40) and 20.4 h (SD: 9.89) for the cases; including 17.7 h (SD: 9.98) and 25.2 h (SD: 9.9; p = 0.06) in those cases with and without an identifiable pathogen, respectively.

## Performance of metagenomics NGS relative to conventional testing

The mean total number of mNGS sequencing reads or the average percentage of non-host reads (reads that did not map to the human genome) did not differ among decedents in whom infection was attributed in the causal pathway to death by the DeCoDe panel, between those with (4.7 million paired-end reads and 13.0%; n = 300 244/2 305 246) and without an identifiable pathogen (5.5 million paired-end reads and 8.4%; n = 161 118/1 905 520); Fig. 1. Overall the controls decedents had lower non-host reads compared with cases who had an infectious CoD (1.8% vs 10.7%; p = 0.08). Furthermore, the host read percentages were significantly lower in the cc, but not so in cases in whom no organism had been identified (14.2%; 336 948/2 370 078; p = 0.6); Fig. 1. In 37.5% (n = 6/16) of the control samples, common NGS contaminates were identified at low concentrations and were removed from further case analysis through the control background subtraction matrix analysis. The suspected contaminates included *Burkholderia*, *Bradyrhizobium, Pseudomonas* and *Staphylococcus*. Furthermore, control samples 2 and 7 were weakly positive (<1% genome coverage) for *E.coli*, control samples 3, 12 and 15 were weekly positive for *Pseudomonas* spp., while control samples 2, 4, and 16 were weekly positive for *Serratia marcescens* and control sample 1 was weekly positive for *A. baumannii*. All of the organisms detected in the controls were below the pathogen detection thresholds and thus removed by the background subtraction matrix.

In total, mNGS identified the same organism in 90% (18) of the 20 cases in which the same pathogen had been attributed as the CoD by DeCoDe; Table 1. In the two cases where mNGS failed to detect the putative etiological agent, *Staphylococcus aureus* (case 19) and *Klebsiella pneumoniae* (Case 20) were attributed as the pathogen by culture and/or NAAT (Supplementary Table 2); albeit the NAAT cycle threshold (Ct) values were high (32.8 and 34.9 for *S. aureus* in the plasma and lung samples, respectively; and 34.8 for the *K. pneumoniae* in the lung). Furthermore, a single death attributed to respiratory syncytial virus (RSV), an RNA virus, which had been identified by NAAT on lung tissue (Ct = 30) was negative on mNGS (plasma and lung). The mNGS detected DNA viruses in three cases, including human alphaherpesvirus 1 (Case 19 in plasma; not previously detected by NAAT), CMV (Case 18 in plasma and lung; previously detected by NAAT in the blood sample), and human adenovirus (Case 14 in plasma; not previously detected by NAAT); Supplementary Table 2. Further, mNGS identified additional organisms which could have contributed to the neonatal deaths, namely *Klebsiella aerogenes* identified concurrently with *A. baumannii* and *S. aureus* (case 8).

Of the 11 cases where death was attributed to an infection, but no organism was identified as the causative agent by the DeCoDe panel (Cases 21–31), potential etiological organisms were identified in nine (81%) decedents using mNGS; Table 1. The putative pathogens identified only by mNGS included *K. pneumoniae* (plasma in case 23), *A. baumannii* (in plasma of case 26 and lung tissue in case 27), and *S. marcescens* (in plasma of cases 30 and 31). In two cases (case 21 and case 28), the same pathogens detected through either culture and/or PCR were also identified by mNGS; Table 1 and Supplementary Table 2.

Furthermore, of the 14 cases that had a positive antemortem blood cultures, mNGS detected the same organisms in 86% (n = 12). Similarly, of the 23 cases that had positive postmortem blood and/or lung tissue culture, mNGS detected the same organisms in 87% (n = 20). Overall, NAAT had a low sensitivity for detecting the etiological CoD (40%, n = 8/20). Similarly, the agreement between NAAT and mNGS was also low (50%; n = 4/8); Table 1 and Supplementary Table 2.

The raw paired-end reads from the mNGS RNA libraries were screened for the presence of AMR genes. In total, 32 AMR genes were identified within the mNGS data (Fig. 2), including resistance genes to beta-lactams, aminoglycosides, sulfonamide, trimethoprim, phenicols,

**Table 1 | Use of mNGS in detection of putative pathogens in neonates in whom infections attributed as the cause of death**

| Case number | Underlying CoD[a] | Immediate CoD[a] | Direct etiologic agent CoD[a] | RNA libraries | | DNA libraries | |
|---|---|---|---|---|---|---|---|
| | | | | Lung metagenomic | Blood metagenomic | Lung metagenomic | Blood metagenomic |
| 1 | Congenital anomalies | LRTI | A.baum | Failed library prep | A.baum | Negative | Negative |
| 2 | Prematurity | Meningitis | A.baum | A.baum, K.pneu | A.baum, Kpneu | Negative | Negative |
| 3 | Sepsis | LRTI | A.baum, RSV | A.baum | A.baum | *A.baum*[b] | Negative |
| 4 | Prematurity | LRTI | A.baum | A.baum, U.urealyticum | A.baum, U.urealyticum | *A.baum*[b], U.urealyticum | *A.baum*[b] |
| 5 | Prematurity | LRTI | A.baum | A.baum | A.baum | *A.baum*[b] | Negative |
| 6 | Prematurity | Sepsis | A.baum, S.aur, ureaplasma | A.baum | A.baum, ureaplasma, S.aur | *A.baum*[b] | *A.baum*[b] |
| 7 | Prematurity | Sepsis | E.coli | Negative | E.coli | Negative | Negative |
| 8 | Prematurity | Sepsis | S.aur, A.baum | A.baum, S.aur, K.aero | A.baum, K.aero | Negative | Negative |
| 9 | Prematurity | Sepsis | S.marc | S.marc | S.marc | Negative | Negative |
| 10 | LRTI | Sepsis | A.baum | A.baum, S.heme | A.baum | *A.baum*[b] | *A.baum*[b] |
| 11 | Prematurity | Sepsis | A.baum | A.baum | A.baum | Negative | Negative |
| 12 | Prematurity | LRTI | A.baum | A.baum | A.baum | *A.baum*[b], E.coli | Negative |
| 13 | Prematurity | LRTI | A.baum | A.baum | A.baum | Not done | Not done |
| 14 | Congenital anomalies | Sepsis | A.baum | A.baum | A.baum, S.marc, Human adenovirus | Not done | Not done |
| 15 | Prematurity | Sepsis | A.baum, K.pneu | A.baum | S.marc, A.baum | Not done | Not done |
| 16 | Prematurity | Sepsis | A.baum, S.aur | Negative | A.baum | Not done | Not done |
| 17 | Prematurity | LRTI | A.baum | A.baum | A.baum, S.marc | Not done | Not done |
| 18 | Prematurity | Sepsis | K.pneu | CMV, S.aur | K.pneu, CMV | Not done | Not done |
| 19 | Prematurity | Sepsis | MRSA | Negative | S.marc, Human alpha-herpesvirus 2 | Not done | Not done |
| 20 | Prematurity | LRTI | K.pneu | Failed library prep | S.marc | Not done | Not done |
| 21 | Prematurity | Sepsis | Unspecified organism | K.pneu | E.coli, K.pneu | K.pneu | Negative |
| 22 | Sepsis | Respiratory distress syndrome | Unspecified organism | Negative | Negative | Negative | Negative |
| 23 | Prematurity | Sepsis | Unspecified organism | Negative | K.pneu, S.mitis | Negative | Negative |
| 24 | Prematurity | Sepsis | Unspecified organism | H.heme | Negative | H.heme | Negative |
| 25 | Prematurity | Congenital pneumonia | Unspecified organism | K.oxy | Negative | Negative | Negative |
| 26 | Prematurity | Sepsis | Unspecified organism | Negative | A.baum | Negative | Negative |
| 27 | Prematurity | Sepsis | Unspecified organism | A.baum | Negative | Not done | Not done |
| 28 | Prematurity | Sepsis | Unspecified organism | Negative | S.aur | Not done | Not done |
| 29 | Prematurity | Sepsis | Unspecified organism | Negative | Negative | Not done | Not done |
| 30 | Prematurity | Sepsis | Unspecified organism | Negative | S.marc | Not done | Not done |
| 31 | Prematurity | Sepsis | Unspecified organism | Negative | S.marc | Not done | Not done |

*LRTI* Lower respiratory tract infection, *A.baum* Acinetobacter baumannii, *K.pneu* Klebsiella pneumonia, *E.coli* Escherichia coli, *RSV* respiratory syncytial virus, *CMV* cytomegalovirus, *S.aur* Staphylococcus aureus, *K.aero* Klebsiella aerogenes, *S.marc* Serratia marcescens, *S.heme* Staphylococcus haemolyticus, *MRSA* Methicillin-resistant Staphylococcus aureus, haemolyticus, *H.heme* Haemophilus haemolyticus, *S.mitis* Streptococcus mitis, *P.mira* Proteus mirabilis, *U.urealyticum* Ureaplasma urealyticum

[a]As determined by the Determination of Cause of Death (DeCoDe) panel of experts who reviewed all available medical and laboratory reports, including post-mortem blood and lung tissue culture, NAAT using Fast Track diagnostic assays on lung and blood samples. All antemortem and postmortem results are reported in Supplementary Table 2.

[b] Indicate where whole genomes were generated

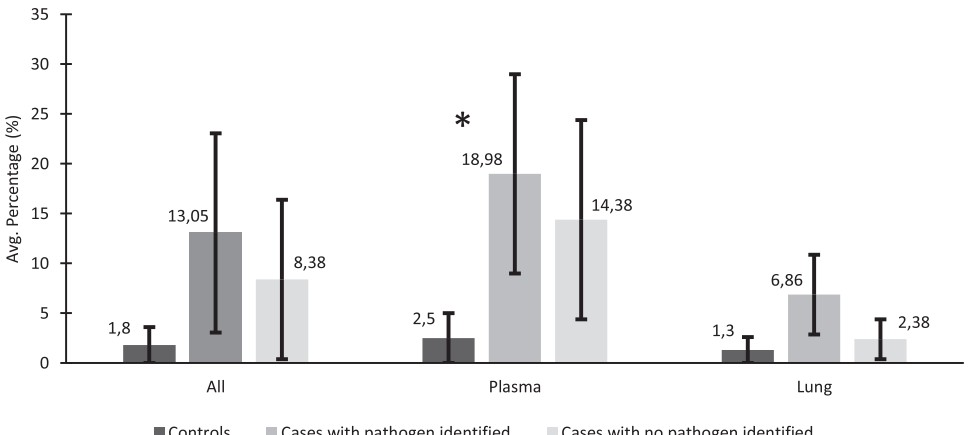

**Fig. 1 | The metagenomic NGS metric for reads remaining after host and quality filtering.** Shows the mean percentage of total reads that were non-human reads overall and for plasma and lung separately from the neonatal infectious disease deaths where a causative pathogen was/was not identified and neonates that died of non-infectious causes. Pathogens were identified through antemortem and postmortem cultures or NAAT. *P* values of <0.05 were considered significant and denoted with an*.

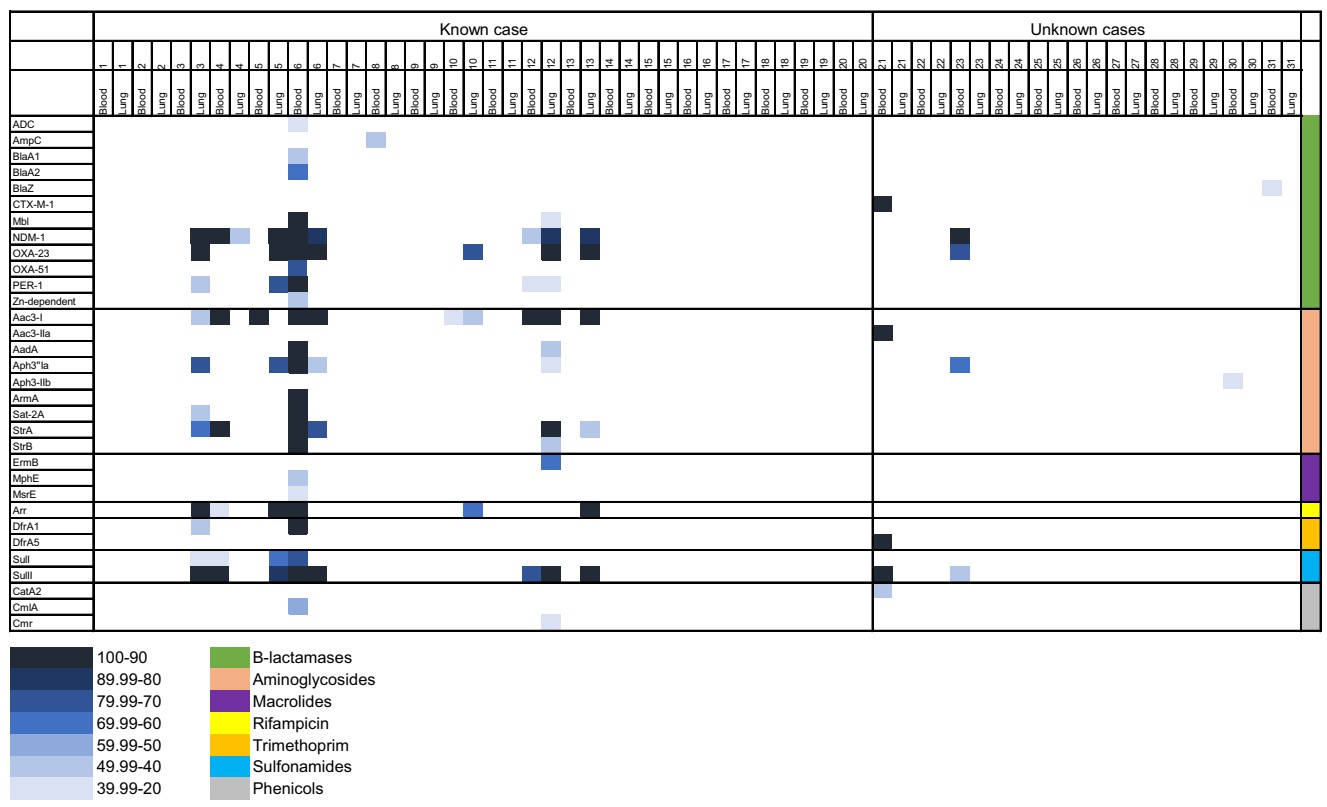

**Fig. 2 | Antimicrobial resistance (AMR) genes in samples from neonates that died of known and unknown pathogens using RNA libraries.** The plot was generated by percentage coverage of the AMR gene and each AMR gene (Y-axis) is organized by antimicrobial class (β-lactamase, aminoglycoside, macrolides, rifamycin, trimethoprim, sulfonamides, and phenicols). Only AMR markers with a coverage >20% and ≥5 reads per sample were included in the analysis.

rifampacin, and macrolides antibiotics. β-lactamase encoding genes exhibited the greatest diversity with 12 unique β-lactamase gene mutations identified, the most common of which was NDM-1 gene (*n* = 7/31, 23%). Furthermore, there were nine different genes indicative of resistance against aminoglycosides, with 32% (10/31) of the cases having at least one aminoglycoside gene detected in the lung and/or blood sample. Sulfonamide and rifamycin resistant genes were each detected in 26% (8/31) of the cases. Of the 32 AMR detected, 56% (18/32) were detected in <5% of the cases, and were mainly related to macrolide, trimethoprim, and phenicol antibiotic resistance.

An overall comparison of the AMR gene abundance revealed that the cases where a pathogen was determined to be the CoD had a greater number of AMR genes (average 2.05, range: 0–23) compared with infectious death cases where no pathogen was identified (average 0.5, range 0–5; *P* = 0.04).

DNA libraries were prepared for 12 of the cases where the infectious agent was identified and six of the cases where no pathogen was identified; Table 1. Amongst the cases with known etiological agents, the DNA library preparations were able to identify the putative organism, in 50% (*n* = 6/12) of the cases in whom an infection

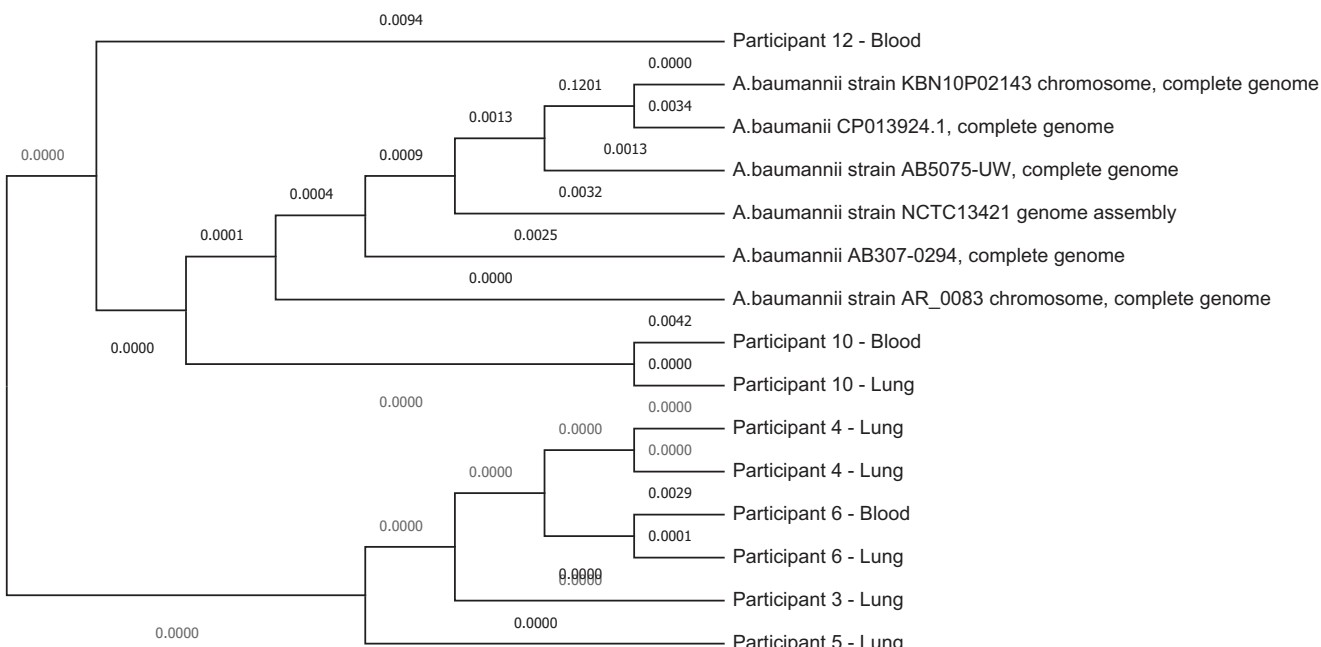

**Fig. 3 | Whole genome phylogeny of the genomes of the 9 *Acinetobacter baumannii* strains collected from 6 neonatal deaths evaluated in the DNA libraries for this study together with 6 reference strains.** The phylogeny tree was constructed on the basis of SNPs and the strains isolated from the same decedent but from different sample type clustered very closely together and all strains clustered together indicating a likely outbreak. The plot was generated using PathogenWatch and the data are available at https://pathogen.watch/collection/urkelmvr3ade-gce-abaumannii-genomes-baillie-et-al-2023.

associated organism was identified by culture or NAAT, all of which were *A. baumannii*. Furthermore, we were able to generate almost complete genome sequences (92–99%) for all six of the *A. baumannii* positive samples, which were all closely related and of ST1 type, Fig. 3. The genome sequences are available on PathogenWatch (https://pathogen.watch/collection/urkelmvr3ade-gce-abaumannii-genomes-baillie-et-al-2023). In the six cases where no pathogen was identified, the DNA library preparations identified a single case of *K. pneumoniae* (16%, *n* = 1/6); Table 1.

The raw paired-end reads from the DNA libraries where an organism was detected were screened for the presence of AMR markers. Similar to the RNA libraries, 29 AMR gene markers were identified, which indicated resistances to beta-lactam, aminoglycosides, sulfonamide, trimethoprim, phenicols, rifampacin, macrolides, and tetracyclines. β-lactamase and aminoglycosides encoding genes showed the greatest diversity with 9 genes identified for each; Fig. 4.

## Discussion

We evaluated the clinical utility of mNGS for identifying putative pathogens in neonates that died due to pneumonia, sepsis and/or meningitis. Metagenomics sequencing was able to corroborate the etiological agent in 90% (*N* = 18/20) of decedents who had a putative pathogen identified by ante- or post-mortem conventional culture or post-mortem NAAT detection in blood or lung samples. In the parent study, hospital-acquired multi-drug resistant bacteria (74%) were the most common pathogens attributed as the CoD, particularly *Acinetobacter baumannii* (52%) despite not being tested for by NAAT[4]. Metagenomic NGS detected *A. baumannii* in all cases where it was identified through conventional culture (100%, *n* = 15/15), and in a further 18% (*n* = 2/11) of cases where no pathogen was identified despite infection being attributed in the causal pathway for death. Further, mNGS identified a possible pathogen in 78% (*n* = 7/9) of deaths attributed to an infectious cause, but in whom no causative organism was identified by the DeCoDe panel based on the ante and post mortem data. Organisms detected by mNGS among the cases with unknown etiologies included *K. pneumoniae, E. coli* and *S. marcescens*, all of which are common causes of hospital acquired infections in neonates[11,12]. The detection of additional organism only by mNGS illustrates that it is likely more sensitive compared with other conventional methods, as well as the advantage of not being biased by prior assumptions of the role of organisms in causal pathway to death[13].

Furthermore, 32 different AMR markers were detected on mNGS associated with resistances to commonly utilized antibiotics[14], including Beta lactamases, macrolides, and aminoglycosides. The identification of antimicrobial resistance genes is important for understanding the evolution and epidemiology of antibiotic resistance and could assist in informing empiric antibiotic treatment. Also, timeous read-out of antibiotic susceptibility profile, should mNGS be deployed on ante-mortem sample testing, could assist in antibiotic stewardship and positively impact on patient outcome with tailoring to appropriate antibiotic sooner than is achievable with conventional culture as antibiotic susceptibility testing which can take up to 6 days[15–18]. Further, due to low blood volumes in neonates (85–100 mL/kg)[19], the volume of blood submitted for culture is often less than the recommended amount (>1 mL) which greatly reduces the sensitivity of blood culture as a diagnostic tool in neonates[20]. In contrast, NGS only requires small sample volumes (~200 μL) and is less affected by prior antibiotic usage unlike conventional culture methods[21]. Yet as mNGS does not always detect whole pathogen genomes the absence of an AMR gene does not imply antibiotic sensitivity. The mNGS antibiotic resistome was however corroborated by the antibiotic susceptibility profile of the pathogens from ante- and post-mortem cultured isolates, where MDR *A. baumannii* was the dominant organism at the time of the study. The findings of the dominance of multi-drug resistance *A. baumannii* as a leading cause of neonatal hospital acquired infections in South Africa is corroborated by earlier studies in South Africa[22–25], and also recognized as a priority organism by the World Health Organization[26].

In this study, both RNA and DNA libraries were generated. Although the RNA libraries were more sensitive for pathogen recovery (90%) compared with the DNA libraries (50%), the latter were able to generate almost complete genome sequences (92–95%) for two-thirds of the organism identified. The near complete genome sequences give

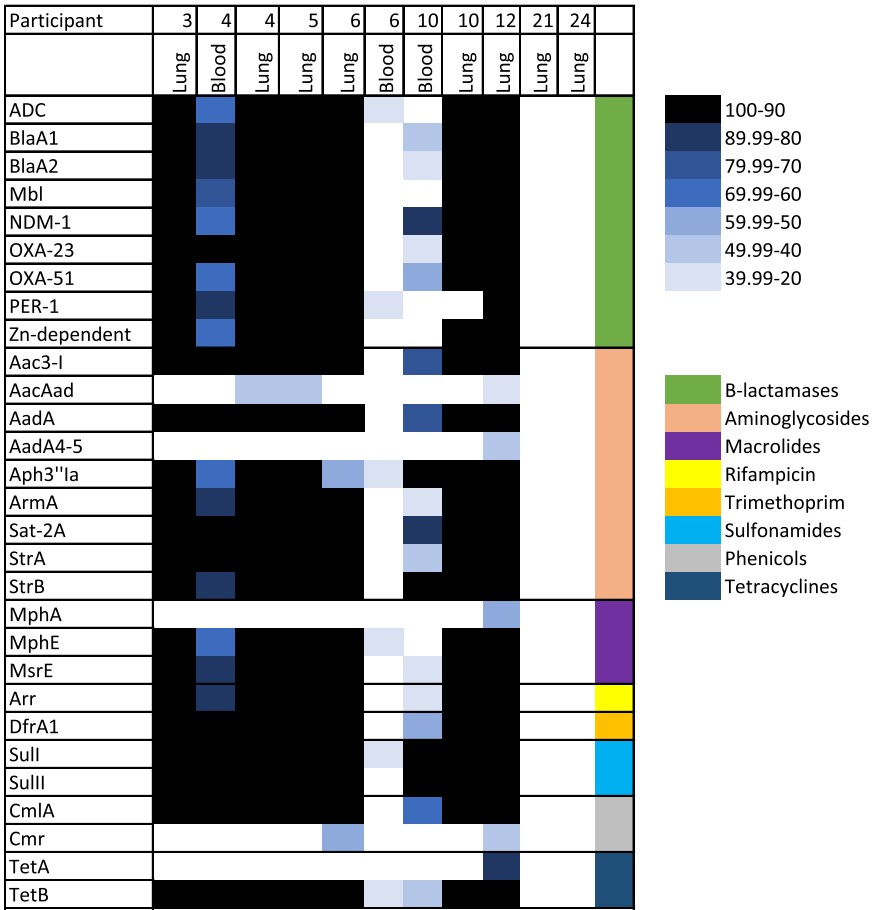

**Fig. 4 | Antimicrobial resistance (AMR) genes in samples from neonates that died of infectious diseases using DNA libraries.** The plot was generated by percentage coverage of the AMR gene and each AMR gene (Y-axis) is organized by antimicrobial class (β-lactamase, aminoglycoside, macrolides, rifamycin, trimethoprim, sulfonamides, phenicols and Tetracyclines). Only samples where an organism was detected and AMR markers with a coverage >20% and ≥5 reads per sample were included in the analysis.

an almost full overview of all the AMR genes present, assisting in being more clinically relevant for tailoring of antibiotic treatment. This was evident by the AMR profiles generated for the DNA libraries which showed resistance genes for all major classes of antibiotics, similar, to the antimicrobial sensitivity culture data. The whole genome sequences could also assist in monitoring for genetic relatedness[27] and identify common source of infection, as alluded to by all the *A. baumannii* isolates that were sequenced being ST1 in our study. An outbreak of *A. baumannii* hospital-acquired infection during our year-long study period is likely to have been the reason for its dominance, as all the decedents had been in the hospital since birth[4]. Thus, DNA metagenomic NGS could be used ante-mortem to define likely transmission events in-hospital and identify putative resistance mutations in emerging strains, which could assist in curbing the transmission and timeously tailoring empiric antibiotic treatment.

Limitations of metagenomic NGS in attributing a causal association to death include the abundance of human host DNA present in the samples (>90% overall, >99% in 49% of samples), which reduced the analytical sensitivity for pathogen recovery. The decedents with an infectious cause of death but where no organism was identified initially, tended to have longer time interval between death and sample collection and lower percentages of nonhost reads (avg: 25.2 h and 2.3%) compared to where a pathogen was implicated (17.7 h and 6.9%). Thus we propose that the postmortem samples where conventional culture and mNGS failed to detect a causative agent were most likely more degraded leading to reduced sensitivity for pathogen detection. Host depletion methods were employed in the RNA libraries to

mitigate the low pathogen recovery. Future studies should investigate additional methods, including CRISPR-Cas9 cleavage, to reduce the host background further[28]. Another limitation of metagenomic NGS is the detection of contaminants present in the sample or introduced during the sequencing process[29,30]. To minimize potential environmental contaminates and commensal flora, negative controls and samples collected from decedents that died of non-infectious causes were sequenced to create a background matrix. Further, the odds of a contaminating or likely colonizing organism (including CMV, *S. mitis* and *H. haemolyticus*) being listed as the etiological agent in the CoD was unlikely as the DeCoDe panel only attributed a pathogen as the CoD after a detailed review of the clinical notes, as well as ante- and post-mortem laboratory findings.

In conclusion, our data shows that clinical metagenomic NGS of samples collected using post-mortem MITS could be useful in determining pathogen-specific infectious causes of neonatal deaths. This diagnostic approach may guide future treatment plans in neonatal intensive care units, help to identify emerging infections and disease phenotypes as well as identify and help to curb nosocomial outbreaks. Further, in resource constrained countries, mNGS could be used to identify the pathogen landscape of a given disease and the most common AMR genes present which could then direct targeted NAAT development.

## Methods

Our research fuls all relevant ethical requirements and was conducted in accordance with the Declaration of Helsinki. The parent study and

the mNGS testing amendment was approved by the Human Research Ethics Committee of the University of the Witwatersrand (HREC reference number 150215). The parents of decedents had provided written informed consent, including for further testing of samples such as undertaken in the current study.

## Study site and population

The samples were collected as part of an observational study aimed to establish the CoD in children under the age of five years in Soweto, South Africa. The detailed characteristics of the study population, study site, MITS procedure, sample testing, and the manner in which the causes of death (CoD) was attributed by a multi-disciplinary panel (referred to as DeCoDe) has been detailed previously[4].

Briefly, from the 16th of July 2015 to the 30th of July 2016, children who died at Chris Hani Baragwanath Academic Hospital in Johannesburg, South Africa were identified by the study staff, and parental consent was obtained for study inclusion. Inclusion criteria included birth weight >750 g, residing in the study site, ability to perform the MITS procedure within 36 h of death, and parental consent. The current study is limited to investigation of deaths that occurred in those <28 days of age (i.e., neonates).

The MITS procedures were conducted by trained study staff—the decedents were washed with water and decontaminated using 70% ethanol prior to any study sampling procedure. Core biopsy needles were used to collect brain, liver, and lung samples. All tissue samples were sent for histopathology examination, culture for bacteria were done on blood and cerebrospinal fluid (CSF), and NAAT were done on lung tissue, blood and CSF samples. The NAAT testing was conducted using commercially available multiplex Fast-Track Diagnostic (FTD™, Malta) PCR assay. The FTD sepsis kit (Cat no: 10921757) was used to test blood samples, and included targets for cytomegalovirus (CMV), Group B *Streptococcus* (GBS), *Listeria monocytogenes, Escherichia coli, Staphylococcus aureus, Chlamydia trachomatis* and *Ureaplasma urealyticum/parvum*. The lung tissue samples were tested using the FTD sepsis, neuro-9 (Cat no: 10921801) and Resp-33 (Cat no: 10921707) PCR panels. In addition to the organism on the FTD sepsis panel, the other organism targets included for detection on lung were epstein-barr virus, adenovirus, herpes simplex virus (1 and 2), varicella-zoster virus, enterovirus, parechovirus, human herpes virus (6 and 7), parvovirus B19, influenza (A, B and C), rhinovirus, coronavirus (NL63, 229E, OC43 and HKU1), parainfluenzae virus (1, 2, 3 and 4), human metapneumovirus, bocavirus, respiratory syncytial virus, *Mycoplasma pneumoniae, Streptococcus pneumoniae, Haemophilus influenzae, Pneumocystis jirovecii, Bordetella* spp., *Moraxella catarrhalis, Klebsiella pneumoniae, Legionella* spp., and *Salmonella* spp.

## Determination of CoD

An international DeCoDe panel consisting of pathologists, pediatricians, epidemiologists, microbiologists, infectious disease specialists, and obstetricians was convened to review all clinical records and antepartum and postmortem testing results to determine the CoD. The characterization of the casual pathway to death included attribution of the "underlying condition" which was considered as the underlying illness leading to death, as well as the final event leading to death; i.e., the "immediate" CoD[31]. In addition, other antecedent conditions were also evaluated.

Cases were decedents in whom an infectious cause of death was attributed by the DeCoDe panel as either an underlying or immediate CoD. The cases were further stratified into those where a specific pathogen was identifiable by the DeCoDe panel as the pathogen contributing to death, and cases where there was insufficient evidence to attribute a pathogen as the CoD even if organisms were detected by culture or NAAT. For example, for sepsis related deaths, there had to be histopathological evidence of sepsis from more than one organ and post-mortem culture and/or positive NAAT for the same organisms in

more than one sample if the ante-mortem culture was negative. Similarly, for LRTI related deaths, there had to be histopathological change of pneumonia and immunohistochemical evidence of the organism in the lung tissue with culture and/or NAAT positivity[4,31].

In addition for a control group, we also undertook mNGS on plasma and lung tissue samples from a group of decedents in whom no notable organisms were detected by culture or NAAT and the DeCoDe panel did not attribute infection related illness in the casual pathway to death. The metagenomic analysis was limited to the decedents with adequate paired blood and lung tissue samples available for testing. We sampled one control for every two case and the investigators were blinded to the case control status as well as the causative agent responsible for the death in the cases.

## Metagenomic analysis

Paired plasma and lung tissue samples were tested using QIAseq FastSelect rRNA/Globin kit (QiaGen, Germany, Cat no: 334375) for human host RNA depletion in line with mNGS library prep. The Ultra II RNA Library Prep Kit (New England Biolabs, Massachusetts, United States, Cat no: E7775) was used according to manufacturer's instruction as previously described[32]. On a subset of samples with sufficient residual material, DNA libraries were generated using the NEBNext Ultra II FS DNA Library Prep kit (New England Biolabs, Massachusetts, United States, Cat no: E7805) according to the manufacturer's instruction. The resulting libraries were sequenced using a 300 cycle High Output Illumina NextSeq500 instrument (Illumina, San Diego, CA, USA). A water control sample was included during each extraction and external RNA controls 103 Consortium collection (ERCC; Life Technologies, Carlsbad, CA, Cat no: 4456740) spike-in controls were included in each sample for the RNA libraries.

## Phylogenetic analysis

The raw fastq files were uploaded to the CZ ID portal which is a cloud-based, open-source metagenomics bioinformatics tool, specifically designed to detect pathogens and antimicrobial resistance markers (AMR) (https://czid.org/version 7.0)[33]. All NGS results and whole genomes are available To remove any potential environmental contaminants and commensal flora, Z-score metric for each genus relative to a background matrix were used where taxa with a Z-score <1 were removed from the analysis. The background matrix was composed of the "no template" water controls and the controls (deaths not attributed to an infectious cause). Organism detections were reported on the basis of additional threshold criteria[34,35], including a reads per million cut-off ≥10, a read alignment length of ≥250 bps and a genome coverage of >5%. All mNGS identified organisms and the genotype for the viruses were confirmed using BLASTn. This analysis was performed separately for the DNA and RNA libraries and the results were compared. For the AMR marker analysis, only genes with a coverage>20% and ≥5 reads per sample were included.

The potential pathogens identified by mNGS were compared with the results from the NAAT and bacteria culture for which the samples had been previously tested. Where the potential organism identified by mNGS was not detected during the initial testing, its presence was confirmed through additional NAAT assays.

## Statistical analysis

Study data were collected and managed using REDCap electronic data capturing tools (Version 13.8.1) which are hosted at the University of the Witwatersrand, Johannesburg, South Africa[36,37]. This is an exploratory study thus no statistical methods were used to pre-determine a sample size and the investigators were blinded to allocation during experiments and outcome assessment. Due to the limited number of participants included in the analysis, sex or gender analysis was not performed and no data was excluded from the analyses. All statistical analysis were conducted using STATA Version 11.0

(StataCorp, Texas, USA) and the statistical significance between groups was calculated using the tests indicated in the figure legends.

## Reporting summary

Further information on research design is available in the Nature Portfolio Reporting Summary linked to this article.

## Data availability

All data needed to evaluate the conclusions of this work are present in the main document and/or the Supplementary Materials. Source data are provided with this paper.

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

## Acknowledgements

This work was supported by the Bill and Melinda Gates foundation OPP1211813 (V.L.B., S.A.M., C.P.O). The authors would like to acknowledge Cristina M. Tato and CZ Biohub Genomics Platform for providing sequencing support and guidance.

## Author contributions

Conceptualization and methodology, supervision, funding acquisition, Data curation, Project administration: V.L.B., S.A.M., C.P.O. Laboratory work: V.L.B., C.P.O., V.A. Data analysis: V.L.B., S.A.M., V.A., C.P.O. Writing of the first draft: V.L.B. Manuscript review and editing: V.L.B., S.A.M., V.A., C.P.O.

## Competing interests

The authors declare no competing interests.
