## [Peer Review file · Nature Communications]

REVIEWER COMMENTS

Reviewer #1 (Remarks to the Author):

Baillie et al. presents on the use of mNGS for detection of putative pathogens from post-mortem sampling of neonates that expired from presumed infection. Although the findings do confirm some value of mNGS in the detection of pathogens directly from samples there are some concerns that are raised that needs further explanation. The following comments are provided for the authors' consideration.

All laboratory findings reported in the manuscript is post-mortem. It would be helpful for the authors to also include lab findings prior to the patients' death to determine if there were any infectious etiologies identified and how it compares to work up performed post-mortem including mNGS. Also would be important to report patients' clinical course including any antimicrobials that the patients were on prior to death. Particularly in the cases of culture/NAAT neg but mNGS pos and in the cases where resistance genes were detected

The *A. baumannii* were found to be closely related – were there concerns for outbreak? Or contamination during the sampling of post-mortem samples? What were the timelines between all the cases positive for *A. baumannii*? How likely can an outbreak vs contamination be ruled out based on this?

The authors raise the concern regarding sample quality and contamination in the discussion but I do think this is a major concern that should not be dismissed lightly. This is where investigating any laboratory findings prior to death would be important. It should be expected that in some of these cases blood cultures or other cultures would have been positive. Correlation with these labs would help corroborate the findings reported in this study. In addition, as there were a high number of *A. baumannii* detected in this study, can the authors share if this is typical with what is normally recovered in the clinical laboratory? Does *A. baumannii* make up a good portion of all pathogens detected from neonates?

Although the study is focused on post-mortem samples one of the premise of the study should be on how mNGS can be utilized to detect pathogens in cases of neonatal sepsis. The authors should discuss this further and what potential utility there is.

There are a lot of run on sentences when the authors are referring to findings from a table. It would be beneficial to separate these into more than one sentence. In its current form it is confusing. For

example: “DNA library preparations were only able to identify the putative organism in 50% (6/12) of the cases in whom an infection associated organism was identified by culture or NAAT (12/20), all *A.baumannii*, and 16% (1/6) of the infectious deaths where no pathogen was identified as the likely CoD (6/11), *K.pneumoniae*; Table 1”

Correct the spelling for *A. baumannii*

Viruses do not need to be capitalized

Abstract: suggest changing “sensitively” to “successfully”

Page 3, 1st paragraph: change “pathogen detection” to “pathogen recovery”

Reviewer #2 (Remarks to the Author):

In this manuscript by Baillie and colleagues, the investigators use mNGS to look for potential pathogens associated with neonatal deaths in Soweto, South Africa, in post-mortem blood and lung tissue samples. The mNGS results were overall very concordant with traditional culture and NAAT, with the same pathogens detected in 90% (18/20) of the decedent. In addition, with culture-/PCR-negative samples, mNGS identified a likely pathogen in 81% (9/11) of cases, all bacteria that are common neonatal pathogens. mNGS was also able to identify an antimicrobial resistance profile associated with each of the cases. The strengths of this study is the identification of a specific clinical use case (neonatal deaths), the availability of paired whole blood and “gold standard” tissue samples, and the high yield of mNGS in both microbiologically confirmed and unknown, culture-/PCR-negative samples. This work would be a welcome contribution to the existing literature on mNGS. However, I have some concerns about the robustness of the study and the analysis of the interpretation, as follows:

Major comments:

(1) Little information is provided on what constituted a detection of a pathogen by mNGS. Is there a reads cutoff or reads per million (RPM) cutoff? Is there a Z-score cutoff after normalization by background. In positive mNGS samples, were additional contaminant/background/co-infecting organism

in addition to the primary organism detected. I am surprised if after normalization the result was detection of the causative organism with no other reads from other organisms.

(2) Related to #2, were any of the pathogens (or reads from that pathogen) that were detected by mNGS and confirmed by culture and PCR detected in control samples? How many control samples were looked at? Using the threshold cutoff for positivity, were any of the control samples positive for any pathogen? Appropriate control samples would be samples from neonates who died of a documented non-infectious etiology (e.g. SIDS, neonatal encephalopathy, preterm birth complications, etc.). It is critical with mNGS that specificity be assessed as well as sensitivity.

(3) For meningitis/encephalitis, cerebrospinal fluid or brain biopsy / autopsy tissue is typically of higher yield than blood or lung tissue. Was either or both sample types tested by culture, PCR, and mNGS?

(4) In the cases associated with unknown deaths, there should have been an a priori suspicion of the cause of death (e.g. bacterial, viral, etc.), even if a causal organism had not been identified. Was the a priori clinical suspicion of CoD consistent with the mNGS result?

(5) Some of the detected pathogens, while certainly credible causes of death, are also associated with colonization and/or contamination so detection alone does not establish these organisms are being the true CoD. For example, *Streptococcus epidermidis* and CMV. Can the authors comment on whether the clinical data support these organisms are being the true CoD? For instance, were blood cultures persistently positive for *Streptococcus epidermidis*, was the CMV viral load high / Ct value low suggesting a high viral pathogen burden?

(6) The significantly greater number of AMR genes in cases where a pathogen was determined may be simply because there are more reads to that pathogen. Can the authors comment on this? Also, it is unclear to me how the AMR gene markers identified would be clinically useful as whole-genome coverage isn't achieved. Thus, the presence of an AMR gene may be helpful but the absence of any given AMR gene does mean that the organism is sensitive. In the absence of full coverage, how useful is the AMR gene analysis if the intent is to run clinical mNGS for identification of AMR in detected organisms. Can the authors compare the AMR genes detected with the known antimicrobial susceptibility results for the positive cultured organisms?

(7) Please clarify whether the blood samples are whole blood or plasma (cell-free metagenomics). Whole blood has higher background, which may explain potential lower yield but enrichment can be obtained by differential lysis of bacteria and fungi, which does not appear to have been done here. Plasma relies on cell-free detection of bacteria, fungi, and viruses and is of lower background so may be more sensitive by mNGS.

REVIEWER COMMENTS

Reviewer #1 (Remarks to the Author):

Baillie et al. presents on the use of mNGS for detection of putative pathogens from post-mortem sampling of neonates that expired from presumed infection. Although the findings do confirm some value of mNGS in the detection of pathogens directly from samples there are some concerns that are raised that needs further explanation. The following comments are provided for the authors' consideration.

All laboratory findings reported in the manuscript is post-mortem. It would be helpful for the authors to also include lab findings prior to the patients' death to determine if there were any infectious etiologies identified and how it compares to work up performed post-mortem including mNGS. Also would be important to report patients' clinical course including any antimicrobials that the patients were on prior to death. Particularly in the cases of culture/NAAT neg but mNGS pos and in the cases where resistance genes were detected.

Thank you for this comment – we have included the antemortem culture findings and antibiotic use prior to death for cases (Supplementary table 2) and controls (Supplementary Table 1). The culture susceptibility and resistance have also been reported for both the ante and postmortem cultures and the agreement between the two have been reported on.

The *A. baumannii* were found to be closely related – were there concerns for outbreak? Or contamination during the sampling of post-mortem samples? What were the timelines between all the cases positive for *A. baumannii*? How likely can an outbreak vs contamination be ruled out based on this?

Hospital acquired infections with *A.baumannii* has been an ongoing challenge at the hospital, in addition to other bacteria such as *K. pneumoniae*. We have discussed this further in the manuscript and have referenced a number of other studies in South Africa that reports an increased detection of multi-drug resistant, nosocomial *A.baumannii* neonatal sepsis.

The authors raise the concern regarding sample quality and contamination in the discussion but I do think this is a major concern that should not be dismissed lightly. This is where investigating any laboratory findings prior to death would be important. It should be expected that in some of these cases blood cultures or other cultures would have been positive. Correlation with these labs would help corroborate the findings reported in this study. In addition, as there were a high number of *A. baumannii* detected in this study, can the authors share if this is typical with what is normally recovered in the clinical laboratory? Does *A. baumannii* make up a good portion of all pathogens detected from neonates?

We have included the antemortem and postmortem culture results and antibiotic resistances; Supp Table 2, and there is good agreement between the antemortem (where positive) and postmortem

culture results. We have amended the discussion to elaborate on this including details on the DeCoDe process which takes into account all tests and clinical records done before the neonate died as well as the lab results from post-mortem testing. We have also elaborated more on the impact that *A. baumannii* has in South Africa and globally as a WHO-defined critical ESKAPE pathogen.

Although the study is focused on post-mortem samples one of the premise of the study should be on how mNGS can be utilized to detect pathogens in cases of neonatal sepsis. The authors should discuss this further and what potential utility there is.

Thank you for this suggestion - we have elaborated in the discussion on the potential advantages of using NGS in diagnosing neonatal sepsis.

There are a lot of run on sentences when the authors are referring to findings from a table. It would be beneficial to separate these into more than one sentence. In its current form it is confusing. For example: "DNA library preparations were only able to identify the putative organism in 50% (6/12) of the cases in whom an infection associated organism was identified by culture or NAAT (12/20), all *A.baumannii*, and 16% (1/6) of the infectious deaths where no pathogen was identified as the likely CoD (6/11), *K.pneumoniae*; Table 1"

Noted, we have shortened the sentences throughout to enhance readability.

Correct the spelling for *A. baumannii* – **Have corrected it throughout.**

Viruses do not need to be capitalized – **Have corrected throughout.**

Abstract: suggest changing "sensitively" to "successfully" - **Have made the suggested change.**

Page 3, 1st paragraph: change "pathogen detection" to "pathogen recovery" – **Have made the suggested changes.**

Reviewer #2 (Remarks to the Author):

In this manuscript by Baillie and colleagues, the investigators use mNGS to look for potential pathogens associated with neonatal deaths in Soweto, South Africa, in post-mortem blood and lung tissue samples. The mNGS results were overall very concordant with traditional culture and NAAT, with the same pathogens detected in 90% (18/20) of the decedent. In addition, with culture-/PCR-negative samples, mNGS identified a likely pathogen in 81% (9/11) of cases, all bacteria that are common neonatal pathogens. mNGS was also able to identify an antimicrobial resistance profile associated with each of the cases. The strengths of this study is the identification of a specific clinical use case (neonatal deaths), the availability of paired whole blood and "gold standard" tissue samples, and the high yield of mNGS in both microbiologically confirmed and unknown, culture-/PCR-negative samples. This work would be a welcome contribution to the existing literature on mNGS. However, I have some concerns about the robustness of the study and the analysis of the interpretation, as follows:

Major comments:

(1) Little information is provided on what constituted a detection of a pathogen by mNGS. Is there a reads cutoff or reads per million (RPM) cutoff? Is there a Z-score cutoff after normalization by background. In positive mNGS samples, were additional contaminant/background/co-infecting organism in addition to the primary organism detected. I am surprised if after normalization the result was detection of the causative organism with no other reads from other organisms.

We have now included the analysis thresholds and references in the methods. The thresholds and background matrix (comprised of the “control” participants and the water controls) cutoff were relatively stringent to allow for only the identification of the CoD pathogen. Further, the blood and lungs are considered sterile sites so we were not expecting high concentrations of other pathogens not related to the cause of death.

(2) Related to #2, were any of the pathogens (or reads from that pathogen) that were detected by mNGS and confirmed by culture and PCR detected in control samples? How many control samples were looked at? Using the threshold cutoff for positivity, were any of the control samples positive for any pathogen? Appropriate control samples would be samples from neonates who died of a documented non-infectious etiology (e.g. SIDS, neonatal encephalopathy, preterm birth complications, etc.). It is critical with mNGS that specificity be assessed as well as sensitivity.

Yes – 16 controls who an expert panel attributed the death to non-infectious causes were included in the study. We have included more details on the controls to highlight their inclusion; Supplementary Table 1. Briefly, these controls were included together with the no template water controls in the background matrix in order to remove any potential environmental contaminants and commensal flora so that potential pathogen microbes could be more easily distinguished.

(3) For meningitis/encephalitis, cerebrospinal fluid or brain biopsy / autopsy tissue is typically of higher yield than blood or lung tissue. Was either or both sample types tested by culture, PCR, and mNGS?

During the entire parent study, only 3% (5/153) of the cases had meningitis/encephalitis included in the CoD pathway. The CSF samples were obtained by puncturing the Cisterna magna and often the fluid obtained was of very small quantities or were severely haemolysed. In the 5 cases that had meningitis/encephalitis included in the CoD pathway, only 1 had residual blood and lung samples available for NGS and none had residual CSF samples. For these reasons, although CSF could have been better samples, we were unable to analyse those samples.

(4) In the cases associated with unknown deaths, there should have been an a priori suspicion of the cause of death (e.g. bacterial, viral, etc.), even if a causal organism had not been identified. Was the a priori clinical suspicion of CoD consistent with the mNGS result?

We have included the physician diagnosed CoD for all cases in Supplementary Table 2 and for the majority of the cases (28/31), the physician diagnosed CoD was also infection related. We have also included the antemortem culture from the cases, which further corroborate the mNGS results.

(5) Some of the detected pathogens, while certainly credible causes of death, are also associated with colonization and/or contamination so detection alone does not establish these organisms are being the true CoD. For example, *Streptococcus epidermidis* and CMV. Can the authors comment on whether the

clinical data support these organisms are being the true CoD? For instance, were blood cultures persistently positive for *Streptococcus epidermidis*, was the CMV viral load high / Ct value low suggesting a high viral pathogen burden?

We have elaborated on the DeCoDe process in the methods which takes into account the ante- and post-mortem culture results, and ante-mortem clinical records. The expert panel used all the available data points to determine if the organism was contributing to the causal pathway of death, and not only based on the detection thereof. Neither CMV or *S. epidermidis* were included in the casual pathway of death among the cases we analysed, despite detection thereof by PCR and NGS. We have included the antemortem and postmortem bacterial culture results (and antibiotic susceptibility profile), for which there is concordance; Supplementary Table 2.

(6) The significantly greater number of AMR genes in cases where a pathogen was determined may be simply because there are more reads to that pathogen. Can the authors comment on this? Also, it is unclear to me how the AMR gene markers identified would be clinically useful as whole-genome coverage isn't achieved. Thus, the presence of an AMR gene may be helpful but the absence of any given AMR gene does mean that the organism is sensitive. In the absence of full coverage, how useful is the AMR gene analysis if the intent is to run clinical mNGS for identification of AMR in detected organisms. Can the authors compare the AMR genes detected with the known antimicrobial susceptibility results for the positive cultured organisms?

We have elaborated on the shortfalls of the RNA libraries with regards to the limitations for AMR marker detection. The DNA libraries were able to generate whole genomes for some of the cases and thus gave a clearer understanding of the AMR profile for the pathogens. We have discussed in more detail in the publication. We also included the antemortem and postmortem culture antibiotic susceptibility profile, which corroborate the large number of AMR genes detected in the samples.

(7) Please clarify whether the blood samples are whole blood or plasma (cell-free metagenomics). Whole blood has higher background, which may explain potential lower yield but enrichment can be obtained by differential lysis of bacteria and fungi, which does not appear to have been done here. Plasma relies on cell-free detection of bacteria, fungi, and viruses and is of lower background so may be more sensitive by mNGS.

We focussed our analysis on plasma samples. We did analyse whole blood and plasma samples comparable, however, the plasma samples had a higher detection rate of pathogens. This has been clarified in the main text

REVIEWERS' COMMENTS

Reviewer #1 (Remarks to the Author):

The authors have addressed all the requested changes. No further comments.

Reviewer #2 (Remarks to the Author):

In this revised manuscript by Baillie, et al., the authors use metagenomic next-generation sequencing (mNGS) to identify putative pathogens associated with neonatal deaths in post-mortem blood and lung tissue samples in Soweto, South Africa. They were able to show that mNGS results corroborated culture and PCR data and an expert clinical consensus in 90% (18 of 20) decedents. Furthermore, mNGS identified a putative pathogen in an additional 9 (81%) of 11 cases where no causative pathogen was identified. The authors also detected AMR genes, and suggest that mNGS may be useful for infectious workup of causes of death in post-mortem samples.

The authors have largely addressed the concerns I raised in my prior review. Specifically, they (1) now include analysis thresholds and references in the methods for reporting of a positive pathogen detection, (2) provided more details on the negative controls, (3) explain why CSF was difficult to obtain and use for mNGS, (4) added the physician diagnosed cause of death, which was quite concordant with the mNGS results, (5) clarified that CMV and *Staphylococcus epidermidis* were not thought to be the cause of death in these cases, (4) provided more detail on the relevance of the AMR gene detection, and (5) clarified that plasma samples were used for the mNGS results and not whole blood. I only have a few additional minor suggestions that would be helpful to the reader and those of us in the mNGS field:

(1) Can the authors explicitly add a comment regarding the likely lack of clinical significance or pathogenicity to the detection of CMV and *Staphylococcus epidermidis*? It is important for readers to recognize that detection of a pathogen by mNGS does not necessarily mean that it is causing the disease, but may be due to colonization or contamination.

(2) There is a typo in "*Shigella flexneri*".

(3) The co-detection of *E. coli* and *S. flexneri* may in fact be detection of a single species (e.g., either *E. coli* or *S. flexneri*). The two genomes can be 80-90% identical so it's unclear to me as to whether mNGS

really differentiated two species. Please see Ragupathi, et al., 2018,
[https://www.ncbi.nlm.nih.gov/pmc/articles/PMC5711669/#:~:text=from Escherichia coli is
challenging,are genetically distant %5B2%5D](https://www.ncbi.nlm.nih.gov/pmc/articles/PMC5711669/#:~:text=from Escherichia coli is challenging,are genetically distant %5B2%5D).

REVIEWER COMMENTS

Reviewer #1 (Remarks to the Author):

The authors have addressed all the requested changes. No further comments.

We would like to thank the reviewer for taking the time to review the manuscript again.

Reviewer #2 (Remarks to the Author):

In this revised manuscript by Baillie, et al., the authors use metagenomic next-generation sequencing (mNGS) to identify putative pathogens associated with neonatal deaths in post-mortem blood and lung tissue samples in Soweto, South Africa. They were able to show that mNGS results corroborated culture and PCR data and an expert clinical consensus in 90% (18 of 20) decedents. Furthermore, mNGS identified a putative pathogen in an additional 9 (81%) of 11 cases where no causative pathogen was identified. The authors also detected AMR genes, and suggest that mNGS may be useful for infectious workup of causes of death in post-mortem samples.

The authors have largely addressed the concerns I raised in my prior review. Specifically, they (1) now include analysis thresholds and references in the methods for reporting of a positive pathogen detection, (2) provided more details on the negative controls, (3) explain why CSF was difficult to obtain and use for mNGS, (4) added the physician diagnosed cause of death, which was quite concordant with the mNGS results, (5) clarified that CMV and *Staphylococcus epidermidis* were not thought to be the cause of death in these cases, (4) provided more detail on the relevance of the AMR gene detection, and (5) clarified that plasma samples were used for the mNGS results and not whole blood. I only have a few additional minor suggestions that would be helpful to the reader and those of us in the mNGS field:

(1) Can the authors explicitly add a comment regarding the likely lack of clinical significance or pathogenicity to the detection of CMV and *Staphylococcus epidermidis*? It is important for readers to recognize that detection of a pathogen by mNGS does not necessarily mean that it is causing the disease, but may be due to colonization or contamination.

Thank you for this comment – We have addressed this in the discussion.

(2) There is a typo in "*Shigella flexneri*".

After reviewing the reviewer's third comment, we believe *S. flexneri* should not have been included and has been removed all together.

(3) The co-detection of *E. coli* and *S. flexneri* may in fact be detection of a single species (e.g., either *E. coli* or *S. flexneri*). The two genomes can be 80-90% identical so it's unclear to me as to whether mNGS really differentiated two species. Please see Ragupathi, et al., 2018, <https://www.ncbi.nlm.nih.gov/pmc/articles/PMC5711669/#:~:text=from Escherichia coli is challenging,are genetically distant %5B2%5D>.

Thank you for picking up this issue. After reviewing this article we agree that is unlikely that these two organisms were detected in a single sample. After reviewing the sequencing data, together with the culture data, the table and text has been updated to reflect that only *E.coli* was detected.